# TRACTABLE PROBABILISTIC GRAPH REPRESENTATION LEARNING WITH GRAPH-INDUCED SUM-PRODUCT NETWORKS

**Federico Errica**
NEC Laboratories Europe
Heidelberg, Germany

**Mathias Niepert**
University of Stuttgart
Stuttgart, Germany

## ABSTRACT

We introduce Graph-Induced Sum-Product Networks (GSPNs), a new probabilistic framework for graph representation learning that can tractably answer probabilistic queries. Inspired by the computational trees induced by vertices in the context of message-passing neural networks, we build hierarchies of sum-product networks (SPNs) where the parameters of a parent SPN are learnable transformations of the a-posterior mixing probabilities of its children's sum units. Due to weight sharing and the tree-shaped computation graphs of GSPNs, we obtain the efficiency and efficacy of deep graph networks with the additional advantages of a probabilistic model. We show the model's competitiveness on scarce supervision scenarios, under missing data, and for graph classification in comparison to popular neural models. We complement the experiments with qualitative analyses on hyper-parameters and the model's ability to answer probabilistic queries.

## 1 INTRODUCTION

As the machine learning field advances towards highly effective models for language, vision, and applications in the sciences and engineering, many practical challenges stand in the way of widespread adoption. Overconfident predictions are hard to trust and most current models are not able to provide uncertainty estimates of their predictions (Pearl, 2009; Hüllermeier & Waegeman, 2021; Mena et al., 2021). Capturing such functionality requires the ability to efficiently answer probabilistic queries, e.g., computing the likelihood or marginals and, therefore, learning tractable distributions (Choi et al., 2020). Such capabilities would not only increase the trustworthiness of learning systems but also allow us to naturally cope with missing information in the input through marginalization, avoiding the use of ad-hoc imputation methods; this is another desideratum in applications where data (labels and attributes) is often incomplete (Dempster et al., 1977; Zio et al., 2007). In numerous application domains, obtaining labeled data is expensive, and while large unlabeled data sets might be available, this does not imply the availability of ground-truth labels. This is the case, for instance, in the medical domains (Tajbakhsh et al., 2020) where the labeling process must comply with privacy regulations and in the chemical domain where one gathers target labels via costly simulations or in-vitro experiments (Hao et al., 2020).

In this work, we are interested in data represented as a graph. Graphs are a useful representational paradigm in a large number of scientific disciplines such as chemistry and biology. The field of Graph Representation Learning (GRL) is concerned with the design of learning approaches that directly model the structural dependencies inherent in graphs (Bronstein et al., 2017; Hamilton et al., 2017b; Zhang et al., 2018b; Wu et al., 2020; Bacciu et al., 2020b). The majority of GRL methods implicitly induce a computational directed acyclic graph (DAG) for each vertex in the input graph, alternating learnable message passing and aggregation steps (Gilmer et al., 2017; Bacciu et al., 2020b). Most of these approaches exclusively rely on neural network components in these computation graphs, but cannot answer probabilistic queries nor exploit the vast amount of unlabeled data.

Motivated by these considerations, we propose a class of hierarchical probabilistic models for graphs called GSPNs, which can tractably answer a class of probabilistic queries of interest and whose computation graphs are also DAGs. While GSPNs are computationally as efficient as Deep Graph

Networks (DGNs), they consist of a hierarchy of interconnected Sum-Product Networks (SPNs) Poon & Domingos (2011). GSPNs can properly marginalize out missing data in graphs and answer probabilistic queries that indicate a change in likelihood under variations of vertex attribute values. The learned probabilistic graph representations are also competitive with state-of-the-art deep graph networks in the scarce supervision and graph classification settings. Overall, we provide evidence that it is possible to build a tractable family of SPNs to tackle learning problems defined on complex, non-Euclidean domains as compositions of simpler ones.

## 2 RELATED WORK

Unsupervised learning for graphs is under-explored relative to the large body of work on supervised graph representation learning (Scarselli et al., 2009; Micheli, 2009; Niepert et al., 2016; Kipf & Welling, 2017; Velickovic et al., 2018; Xu et al., 2019; Ying et al., 2021). Contrary to self-supervised pre-training (Hu et al., 2020b) which investigates ad-hoc learning objectives, unsupervised learning encompasses a broader class of models that extract patterns from unlabeled data. Most unsupervised approaches for graphs currently rely on *i)* auto-encoders, such as the Graph Auto-Encoder (GAE) (Kipf & Welling, 2016) and *ii)* contrastive learning, with Deep Graph Infomax (DGI) (Velickovic et al., 2019) adapting ideas from computer vision and information theory to graph-structured data. While the former learns to reconstruct edges, the latter compares the input graph against its corrupted version and learns to produce different representations.

Existing probabilistic approaches to unsupervised deep learning on graphs often deal with clustering and Probability Density Estimation (PDE) problems; a classic example is the Gaussian Mixture Model (GMM) (Bishop, 2006) capturing multi-modal distributions of Euclidean data. The field of Statistical Relational Learning (SRL) considers domains where we require both uncertainty and complex relations; ideas from SRL have recently led to new variational frameworks for (un-)supervised vertex classification (Qu et al., 2019). Other works (Zheng et al., 2018) decompose the original graph into sub-graphs that are isomorphic to pre-defined templates to solve node-classification tasks. The Contextual Graph Markov Model (CGMM) (Bacciu et al., 2020a) and its variants (Atzeni et al., 2021; Castellana et al., 2022) are unsupervised DGNs trained incrementally, i.e., layer after layer, which have been successfully applied to graph classification tasks. Their incremental training grants closed-form learning equations at each layer, but it comes at the price of no global cooperation towards the optimization of the learning objective, i.e., the likelihood of the data; this makes it impractical for answering missing data queries.

So far, the literature on missing data in graphs, i.e., using the graph structure to handle missing attribute values, has not received as much attention as other problems. Recent work focuses on mitigating "missingness" in vertex classification tasks (Rossi et al., 2022) or representing missing information as a learned vector of parameters (Malone et al., 2021), but the *quality* of the learned data distribution has not been discussed so far (to the best of our knowledge). There are some attempts at imputing vertex attributes through a GMM (Taguchi et al., 2021), but such a process does not take into account the available graph structure. Similarly, other proposals (Chen et al., 2022) deal with the imputation of vertices having either all or none of their attributes missing, a rather unrealistic assumption for most cases. In this work, we propose an approach that captures the data distribution under missing vertex attribute values and requires no imputation.

## 3 BACKGROUND

We define a graph $g$ as a triple $(\mathcal{V}, \mathcal{E}, \mathcal{X})$, where $\mathcal{V} = \{1, \ldots, N\}$ denotes the set of $N$ vertices, $\mathcal{E}$ is the set of *directed* edges $(u, v)$ connecting vertex $u$ to $v$, and $\mathcal{X} = \{\boldsymbol{x}_u \in \mathbb{R}^d, d \in \mathbb{N}, \forall u \in \mathcal{V}\}$ represents the the set of vertex attributes. When an edge is undirected, it is trasformed into two opposite directed edges (Bacciu et al., 2020b). In this work, we do not consider edge attributes. The *neighborhood* of a vertex $v$ is the set $\mathcal{N}_v = \{u \in \mathcal{V} \mid (u, v) \in \mathcal{E}\}$. Also, access to the $i$-th component of a vector $\boldsymbol{x}$ shall be denoted by $\boldsymbol{x}(i)$ and that of a function $f$'s output as $f(\cdot)_i$.

**Graph Representation Learning** Learning on graph-structured data typically means that one seeks a mapping from an input graph to vertex embeddings (Frasconi et al., 1998), and such a mapping should be able to deal with graphs of varying topology. In general, a vertex $v$'s representation is

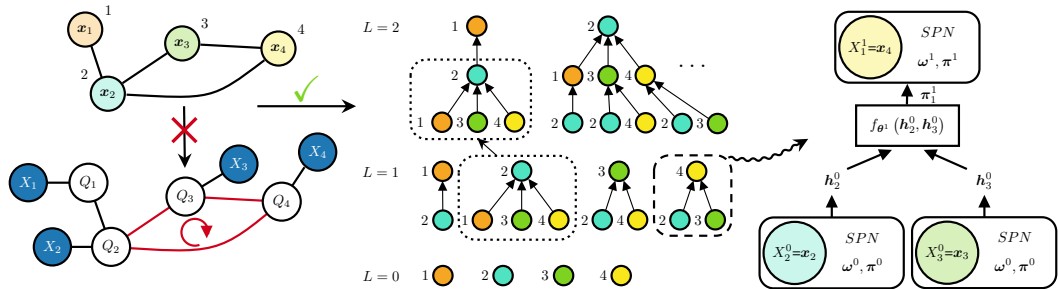

Figure 1: Inference on the graphical model on the left is typically unfeasible due to the mutual dependencies induced by cycles. Therefore, we approximate the learning problem using probabilistic computational trees of height $L$ modeled by a hierarchy of tractable SPNs (right). Note that trees of height $L-1$ are used in the construction process of trees of height $L$. Also, for each tree rooted at $v$ we visualize the mapping $m_v(\cdot)$ using colors and indices corresponding to the original graph (left).

a vector $\boldsymbol{h}_v \in \mathbb{R}^C, C \in \mathbb{N}$ encoding information about the vertex and its neighboring context, so that predictions about $v$ can be then made by feeding $\boldsymbol{h}_v$ into a classical ML predictor. In graph classification or regression, instead, these vertex representations have to be globally aggregated using permutation invariant operators such as the sum or mean, resulting in a single representation $\boldsymbol{h}_g \in \mathbb{R}^C$ of the entire graph that is used by the subsequent predictor. At the time of this writing, message-passing neural networks (MPNN) (Gilmer et al., 2017) are the most popular class of methods to compute vertex representations. These methods adopt a local and iterative processing of information, in which each vertex repeatedly receives messages from its incoming connections, aggregates these messages, and sends new messages along the outgoing edges. Researchers have developed several variants of this scheme, starting from the two pioneering methods: the *recurrent* Graph Neural Network (GNN) of Scarselli et al. (2009) and the *feedforward* Neural Network for Graphs (NN4G) of Micheli (2009).

**Tractable Probabilistic Models** Probabilistic Circuits (PCs) are probabilistic models that can tractably, i.e., with *polynomial* complexity in the size of the circuit, answer a large class of probabilistic queries (Choi et al., 2020) and can be trained via backpropagation. A PC is usually composed of *distribution units*, representing distributions over one or more random variables, *product units* computing the fully factorized joint distribution of its children, and the *sum units* encoding mixtures of the children's distributions. Sum-Product Networks (SPNs) (Poon & Domingos, 2011; Gens & Domingos, 2012; Trapp et al., 2019; Shao et al., 2020) are a special class of PCs that support tractable computation of joint, marginal, and conditional distributions, which we will exploit to easily handle missing data. Informally, a (locally normalized) SPN is a probabilistic model defined via a rooted computational DAG, whose parameters of every sum unit add up to 1 (Vergari et al., 2019a). Furthermore, the *scope* of an SPN is the set of its distribution units, and *valid* SPNs represent proper distributions. For instance, a GMM and a Naïve Bayes model (NB) (Webb et al., 2010) can both be written as SPNs with a single sum unit. While the class of PCs is equivalent to that of deep mixture models, where each sum unit is associated with some latent variable (Peharz et al., 2016), PCs are *not* probabilistic graphical models (PGMs) (Koller & Friedman, 2009): the former specify an operational semantics describing how to compute probabilities, whereas the representational semantics of the latter specifies the conditional independence of the variables involved. Crucially, for valid SPNs and an arbitrary sum unit $j$ with $C$ children, it is always possible to tractably compute its posterior distribution, parametrized by the vector $\boldsymbol{h}_j \in \mathbb{R}^C$ (Peharz et al., 2016). Here, we are voluntarily abusing the notation because in the following we will consider posterior distributions of sum nodes as our latent representations. Due to space constraints, we provide a more detailed introduction to PCs in Section A.1.

## 4  GSPN: LEARNING TRACTABLE PROBABILISTIC GRAPH REPRESENTATIONS

When one considers a cyclic graphical model such as the one on the left-hand-side of Figure 1, probabilistic inference is computationally infeasible unless we make specific assumptions to break

the mutual dependencies between the random variables (*r.v.*) $Q$, depicted as white circles. To address this issue, we approximate the intractable joint probability distribution over the *r.v.* attributed graph of any shape as products of tractable conditional distributions, one for each vertex in the graph; this approximation is known as the pseudo likelihood (Gong & Samaniego, 1981; Liang & Yu, 2003). The scope of these tractable distributions consists of the $L$-hop neighborhoods induced by a traversal of the graph rooted at said vertex. Akin to what done for DGNs, the parameters of the conditional distributions are shared across vertices.

We propose a class of models whose ability to tractably answer queries stems from a hierarchical composition of sum-product networks (SPNs). To describe the construction of this hierarchical model we consider, for each vertex $v$ in the input graph, a tree rooted at $v$ of height $L$, that is the length of the longest downward path to a leaf from $v$, treated as a hyper-parameter. Analogously, an internal node in the tree is said to have height $0 \leq \ell \leq L$. To distinguish between graphs and trees, we use the terms *vertices* for the graphs and *nodes* for the trees. Moreover, because graph cycles induce repetitions in the computational trees, we have to use a new indexing system: given a tree rooted at $v$ with $T_v$ nodes $\mathcal{T} = \{n_1, \ldots, n_{T_v}\}$, we denote by $m_v(\cdot) : \mathcal{T} \to \mathcal{V}$ a mapping from its node index $n$ to a vertex $u$ in the input graph $g$.

We now formally define the tree of height $L$ rooted at vertex $v$ as follows. First, we have one root node $n$ with $m_v(n) = v$ and $n$ having height $L$. Second, for a node $n$ in the tree at height $1 \leq \ell \leq L$, we have that a node $n'$ is a child of $n$ with height $\ell - 1$ if and only if vertex $v' = m_v(n')$ is in the 1-hop neighborhood of vertex $u = m_v(n)$. When vertex $u$ does not have incoming edges, we model it as a leaf node of the tree. Finally, every node $n$ at height $\ell = 0$ is a leaf node and has no children. Figure 1 (center) depicts examples of trees induced by the nodes of the graph on the left.

We use the structure of each graph-induced tree as a blueprint to create a hierarchy of normalized SPNs, where all SPNs have the same internal structure. Every node of a tree is associated with a valid SPN whose scope consists of the random variables $A_1, \ldots, A_d$ modeling the vertex attributes, meaning the distribution units of said SPN are tractable distributions for the variable $\boldsymbol{X} = (A_1, ..., A_d)$. To distinguish between the various distributions for the *r.v.* $\boldsymbol{X}$ modeled by the SPNs at different nodes, we introduce for every node $n$ and every height $\ell$, the *r.v.* $\boldsymbol{X}_n^\ell$ with realization $\boldsymbol{x}_{m(n)}$. Crucially, these *r.v.* are fundamentally different even though they might have the same realization of the *r.v.* for root node . This is because each node of the tree encodes the contextual information of a vertex after a specific number of message passing steps.

The parameters of the tractable distribution units in the SPN of node $n$ at height $\ell$ are denoted by $\boldsymbol{\omega}_n^\ell$, which are shared across the nodes at the same height. Moreover, the mixture probabilities of the $S$ sum nodes in the SPN of node $n$ at height $\ell$ are denoted by $\boldsymbol{\pi}_{n,j}^\ell, 1 \leq j \leq S$. To obtain a hierarchical model, that is, to connect the SPNs according to the tree structure, we proceed as follows. For every SPN of a non-leaf node $n$, the parameters $\boldsymbol{\pi}_{n,j}^\ell$ of sum unit $j$ are learnable transformations of the posterior probabilities of the (latent mixture component variables of) sum units $j$ in the children SPNs. More formally, let $n_1, \ldots, n_T$ be the children of a node $n$ and let $\boldsymbol{h}_{n_i,j}^{\ell-1}$ be the vector of posterior probabilities of sum unit $j$ for the SPN of node $n_i$. Then, $\boldsymbol{\pi}_{n,j}^\ell = f_{\boldsymbol{\theta}_j^\ell}(\boldsymbol{h}_{n_1,j}^{\ell-1}, \ldots, \boldsymbol{h}_{n_T,j}^{\ell-1})$ with learnable parameters $\boldsymbol{\theta}_j^\ell$ shared across level $\ell$. The choice of $f$ does not depend on the specific SPN template, but it has to be permutation invariant because it acts as the neighborhood aggregation function of message-passing methods. The parameters of the sum units of leaf nodes, $\boldsymbol{\pi}_j^0$, are learnable and shared. Figure 1 (right) illustrates the hierarchical composition of SPNs according to a tree in Figure 1 (center). Moreover, Figure 2 illustrates how the prior distribution of the SPNs at height 1 is parametrized by a learnable transformation of posterior mixture probabilities of its children SPNs, similar to what is done by Vergari et al. (2019b) with fixed Dirichlet priors.

A graph $g$ in the training data specifies (partial) evidence $\boldsymbol{x}_v$ for every one of its vertices $v$. The hierarchical SPN generated for $v$ now defines a tractable probability distribution for the root node conditioned on the (partial) evidence for its intermediate nodes, which we can compute by evaluating the hierarchical SPNs tree in a bottom-up fashion. For each vertex $v$ in the graph $g$, let $n_2, \ldots, n_{T_v}$ be the nodes of the graph-induced tree rooted at $v$, without the root node $n_1$ itself. For each graph $g$ in the dataset, the objective is to maximize the following pseudo log-likelihood:

$$\log \prod_{v \in \mathcal{V}} P_{\boldsymbol{\Theta}}(\boldsymbol{x}_{m_v(n_1)} \mid \boldsymbol{x}_{m_v(n_2)}, \ldots, \boldsymbol{x}_{m_v(n_{T_v})}), \tag{1}$$

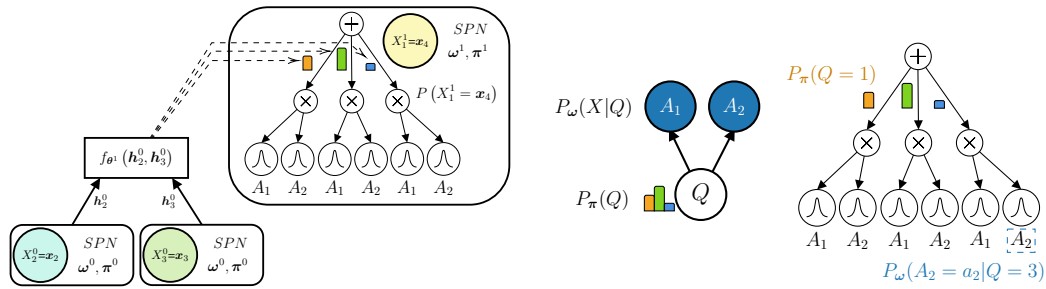

Figure 2: (Left) We expand the example of Figure 1 to illustrate how the prior distribution of the top SPN (here the Naïve Bayes on the right) is parametrized by a learnable transformation of the children's SPNs posterior mixture probabilities. (Right) A Gaussian Naïve Bayes graphical model with *r.v.* $\boldsymbol{X} = (A_1, A_2)$ and its equivalent SPN with scope $\{A_1, A_2\}$.

where $P_{\boldsymbol{\Theta}}$ is the conditional probability defined by the hierarchical SPN for the tree rooted at $v$ with parameters $\boldsymbol{\Theta}$. When clear from the context, in the following we omit the subscript $v$ from $m_v(\cdot)$.

## 4.1 NAIVE BAYES GSPNs

An instance of the proposed framework uses Naïve Bayes models as base SPNs, shown in Figure 2 in both their graphical and SPN representations for two continuous vertex attributes. For each SPN, there is a single categorical latent *r.v.* $Q$ with $C$ possible states. Let us denote this latent *r.v.* for node $n$ at height $\ell$ in the tree as $Q_n^\ell$ and its prior distribution as $P_{\boldsymbol{\pi}_n^\ell}(Q_n^\ell)$. Due to the assumption that all *r.v.s* $\boldsymbol{X}_n^\ell$ have tractable distributions, we have, for all $\ell, n$ and $i$, that the conditional distribution $P_{\boldsymbol{\omega}_n^\ell}(\boldsymbol{X}_n^\ell \mid Q_n^\ell = i)$ is tractable. Moreover, for each child node $n$ at height $\ell$ we have that

$$\boldsymbol{h}_n^\ell(i) = P(Q_n^\ell = i \mid \boldsymbol{X}_n^\ell = \boldsymbol{x}_{m(n)}). \tag{2}$$

For each leaf $n$ of a tree, the posterior probabilities for the SPNs sum unit are given by

$$\boldsymbol{h}_n^0(i) = P(Q_n^0 = i \mid \boldsymbol{X}_n^0 = \boldsymbol{x}_{m(n)}) = \frac{P_{\boldsymbol{\omega}_n^0}(\boldsymbol{x}_{m(n)} \mid Q_n^0 = i)P_{\boldsymbol{\pi}_n^0}(Q_n^0 = i)}{\sum_{i'}^C P_{\boldsymbol{\omega}_n^0}(\boldsymbol{x}_{m(n)} \mid Q_n^0 = i')P_{\boldsymbol{\pi}_n^0}(Q_n^0 = i')}, \tag{3}$$

where $\boldsymbol{\pi}_n^0$ is learned and shared across leaf nodes. For $\ell \geq 1$, the latent priors $P_{\boldsymbol{\pi}_n^\ell}(Q_n^\ell)$ are parametrized by the output of a learnable transformation of the posterior probabilities of the sum units of the child SPNs. More formally, for node $n'$ at height $\ell + 1, \ell \geq 0$, and with children $ch(n')$ we compute the prior probabilities $\boldsymbol{\pi}_{n'}^{\ell+1}$ as

$$\boldsymbol{\pi}_{n'}^{\ell+1} = f_{\boldsymbol{\theta}^{\ell+1}}(\boldsymbol{h}_1^\ell, ..., \boldsymbol{h}_{|ch_{n'}|}^\ell), \qquad \boldsymbol{h}_n^\ell(i) = \frac{P_{\boldsymbol{\omega}_n^\ell}(\boldsymbol{x}_{m(n)} \mid Q_n^\ell = i)P_{\boldsymbol{\pi}_n^\ell}(Q_n^\ell = i)}{\sum_{i'}^C P_{\boldsymbol{\omega}_n^\ell}(\boldsymbol{x}_{m(n)} \mid Q_n^\ell = i')P_{\boldsymbol{\pi}_n^\ell}(Q_n^\ell = i')}, \tag{4}$$

noting that the posterior $\boldsymbol{h}_n^\ell$ is tractable as the quantities of interest are obtained with a single backward pass in the SPN (Peharz et al., 2016). Figure 2 visualizes how $f_{\boldsymbol{\theta}^1}$ acts on the prior probabilities in the example of Figure 1. In the experiments, we define $f_{\boldsymbol{\theta}^{\ell+1}}(\boldsymbol{h}_1^\ell, ..., \boldsymbol{h}_{|ch_{n'}|}^\ell) = \frac{1}{|ch_{n'}|}\sum_{n \in ch_{n'}} \boldsymbol{\theta}^{\ell+1}\boldsymbol{h}_n^\ell$ for a learnable transition matrix $\boldsymbol{\theta}^{\ell+1} \in \mathbb{R}^{C \times C}$ that specifies how much a child's (soft) state $k$ contributes to the weight of state $i$ in the new prior distribution $P_{\boldsymbol{\pi}_{n'}^{\ell+1}}(Q_{n'}^{\ell+1} = i)$. Each row $k$ of $\boldsymbol{\theta}^{\ell+1}$ must specify a valid probability over $C$ possible states, enforced through normalization techniques. The function is motivated by the observation that it corresponds to applying the "Switching Parent" decomposition to the conditional mixture model defined in (Bacciu et al., 2020a) (deferred to Section A.2 due to space constraints). Likewise, we show in Section A.3 that Equation 4 produces a valid parametrization for the new prior distribution.

This iterative process can be efficiently implemented in the exact same way message passing is implemented in neural networks, because the computation for trees of height $\ell$ can be reused in trees of height $\ell + 1$. In practice, the height $L$ of the computational tree corresponds to the number of layers in classical message passing. Furthermore, our choice of $f_{\boldsymbol{\theta}^\ell}$ leads to a fully probabilistic formulation and interpretation of end-to-end message passing on graphs. Motivated by studies on

the application of gradient ascent for mixture models and SPNs (Xu & Jordan, 1996; Sharir et al., 2016; Peharz et al., 2016; Gepperth & Pfülb, 2021), we maximize Equation 1 with backpropagation. We can execute all probabilistic operations on GPU to speed up the computation, whose overall complexity is linear in the number of edges as in most DGNs. For the interested reader, Section A.4 describes how GSPN can be applied to more general SPNs with multiple sum units and provides the pseudocode for the general inference phase; similarly, Section A.5 proposes a probabilistic shortcut mechanism akin to residual connections (Srivastava et al., 2015; He et al., 2016) in neural networks that mitigates the classical issues of training very deep networks via backpropagation. Finally, we refer to the unsupervised version of GSPN as $\text{GSPN}_U$.

## 4.2 MODELING MISSING DATA

With GSPNs we can deal with partial evidence of the graph, or missing attribute values, in a probabilistic manner. This is a distinctive characteristic of our proposal compared to previous probabilistic methods for vectors, as GSPNs can tractably answer probabilistic queries on graphs by also leveraging their structural information. Similarly, typical neural message passing models deal with missing data by imputing its values *before* processing the graph (Taguchi et al., 2021), whereas GSPN takes the partial evidence into account while processing the graph structure. We take inspiration from the EM algorithm for missing data (Hunt & Jorgensen, 2003): in particular, let us consider a multivariate *r.v.* of a root node $\boldsymbol{X}$ (dropping other indices for ease of notation) as a tuple of observed and missing sets of variables, i.e., $\boldsymbol{X} = (\boldsymbol{X}^{obs}, \boldsymbol{X}^{mis})$. When computing the posterior probabilities of sum nodes $\boldsymbol{h}$, we have to modify Equations 3 and 4 to only account for $\boldsymbol{X}^{obs}$; in SPNs, this equals to setting the distribution units of the missing attributes to 1 when computing marginals, causing the missing variables to be marginalized out. This can be computed efficiently due to the *decomposability* property of the SPNs used in this work (Darwiche, 2003). Additionally, if we wanted to impute missing attributes for a vertex $v$, we could apply the conditional mean imputation formula of Zio et al. (2007) to the corresponding root *r.v.* of the tree $\boldsymbol{X}_1^L = (\boldsymbol{X}_1^{obs}, \boldsymbol{X}_1^{mis})$, which, for the specific case of NB, translates to $\boldsymbol{x}_v^{mis} = \sum_i^C \mathbb{E}\big[\boldsymbol{X}_1^{mis} \mid Q_1^L = i\big] * \boldsymbol{h}_1^L(i)$. Hence, to impute the missing attributes of $\boldsymbol{x}_v$, we sum the (average) predictions of each mixture in the top SPN, where the mixing weights are given by the posterior distribution. The reason is that the posterior distribution carries information about our beliefs after observing $\boldsymbol{X}_v^{obs}$.

## 4.3 A GLOBAL READOUT FOR SUPERVISED LEARNING

Using similar arguments as before, we can build a probabilistic and supervised extension of GSPN for graph regression and classification tasks. It is sufficient to consider a new tree where the root node $r$ is associated with the target *r.v.* $Y$ and the children are all possible trees of different heights rooted at the $N$ vertices of graph $g$. Then, we build an SPN for $r$ whose sum unit $j$ is parametrized by a learnable tansformation $\boldsymbol{\pi}_{r,j} = f_{\boldsymbol{\vartheta}_j}(\cdot)$. This function receives the set $\boldsymbol{h}_{\mathcal{V},j}$ of outputs $\boldsymbol{h}_{u,j}^\ell$ associated with the *top* SPN related to the graph-induced tree of height $\ell$ rooted at $u$, for $0 \leq \ell \leq L$. In other words, this is equivalent to consider all vertex representations computed by a DGN at different layers. For NB models with one sum unit, given (partial) evidence $\boldsymbol{x}_1, \ldots, \boldsymbol{x}_N$ for all nodes, we write

$$P(\boldsymbol{y} \mid \boldsymbol{x}_1, \ldots, \boldsymbol{x}_N) = \sum_{i=1}^{C_g} P_{\boldsymbol{\omega}_r}(\boldsymbol{y} \mid Q = i) P_{\boldsymbol{\pi}_r}(Q = i), \;\; \boldsymbol{\pi}_r \stackrel{\text{def}}{=} f_{\boldsymbol{\vartheta}}(\boldsymbol{h}_{\mathcal{V}}) = \Omega\big(\sum_{u \in \mathcal{V}} \sum_{\ell=1}^{L} \boldsymbol{\vartheta}^\ell \boldsymbol{h}_u^\ell\big),$$

$$(5)$$

where $\boldsymbol{\omega}_r$ and $\boldsymbol{\pi}_r$ are the parameters of the NB and $Q$ is a latent categorical variable with $C_g$ states. Computing $\boldsymbol{\pi}_r$ essentially corresponds to a global pooling operation, where $\boldsymbol{\vartheta}^\ell \in \mathbb{R}^{C_g \times C}$ is another transition matrix and $\Omega$ can be $\frac{1}{LN}$ (resp. the softmax function) for mean (resp. sum) global pooling. We treat the choice of $\Omega(\cdot)$ as a hyper-parameter, and the resulting model is called $\text{GSPN}_S$.

## 4.4 LIMITATIONS AND FUTURE DIRECTIONS

Due to the problem of modeling an inherently intractable probability distribution defined over a cyclic graph, GSPNs rely on a composition of locally valid SPNs to tractably answer probabilistic queries. At training time the realization of the observable *r.v.* of vertex $v$ at the root might be conditioned on the *same* realization of a *different* observable *r.v.* in the tree, which is meant to capture the mutual

dependency induced by a cycle. To avoid conditioning on the same realization, which could slightly bias the pseudo-likelihood in some corner cases, future work might consider an alternative, albeit expensive, training scheme where, for each node and observable attribute, a different computational tree is created and the internal nodes $n$ with $m(n) = v$ are marginalized out.

From the point of view of expressiveness in distinguishing non-isomorphic graphs, past results tell us that more powerful aggregations would be possible if we explored a different neighborhood aggregation scheme (Xu et al., 2019). In this work, we keep the neighborhood aggregation functions $f_{\boldsymbol{\theta}}$ as simple as possible to get a fully probabilistic model with valid and more interpretable probabilities. Finally, GSPN is currently unable to model edge types, which would enable us to capture a broader class of graphs.

## 5    EXPERIMENTS

We consider three different classes of experiments as described in the next sections. Code and data to reproduce the results is available at the following link: `https://github.com/nec-research/graph-sum-product-networks`.

**Scarce Supervision**    Akin to Erhan et al. (2010) for non-structured data, we show that *unsupervised* learning can be very helpful in the scarce supervision scenario by exploiting large amounts of unlabeled data. We select seven chemical graph regression problems, namely *benzene*, *ethanol*, *naphthalene*, *salicylic acid*, *toluene*, *malonaldehyde* and *uracil* (Chmiela et al., 2017), using categorical atom types as attributes, and *ogbg-molpcba*, an imbalanced multi-label classification task (Hu et al., 2020a). Given the size of the datasets, we perform a hold-out risk assessment procedure (90% training, 10% test) with internal hold-out model selection (using 10% of the training data for validation) and a final re-training of the selected configuration. In the case of *ogbg-molpcba*, we use the publicly available data splits. Mean Average Error (MAE) is the main evaluation metric for all chemical tasks except for *ogbg-molpcba* (Average Precision – AP – averaged across tasks). We first train a $\text{GSPN}_U$ on the whole training set to produce unsupervised vertex embeddings (obtained through a concatenation across layers), and then we fit a DeepSets (DS) (Zaheer et al., 2017) classifier on just $0.1\%$ of the supervised samples using the learned embeddings (referred as $\text{GSPN}_{U+DS}$). We do not include $\text{GSPN}_S$ in the analysis because, as expected, the scarce supervision led to training instability during preliminary experiments.
We compare against two effective and well-known unsupervised models, the Graph Auto-Encoder (GAE) and Deep Graph Infomax (DGI). These models have different unsupervised objectives: the former is trained to reconstruct the adjacency matrix whereas the second is trained with a contrastive learning criterion. Comparing $\text{GSPN}_U$, which maximizes the vertices' pseudo log-likelihood, against these baselines will shed light on the practical usefulness of our learning strategy. In addition, we consider the GIN model (Xu et al., 2019) trained exclusively on the labeled training samples as it is one of the most popular DGNs in the literature. We summarize the space of hyper-parameter configurations for all models in Table 4.

**Modeling the Data Distribution under Partial Evidence**    We analyze GSPN's ability at modeling the missing data distribution on real-world datasets. For each vertex, we randomly mask a proportion of the attributes given by a sample from a Gamma distribution with concentration 1.5 and rate $1/2$). We do so to simulate a plausible scenario in which many vertices have few attributes missing and few vertices have many missing attributes. We train all models on the observed attributes only, and then we compute the negative log-likelihood of Equation 1 (NLL) for the masked attributes to understand how good each method is at modeling the missing data distribution. We apply the same evaluation, data split strategy, and datasets of the previous experiment, focusing on their 6 continuous attributes (except for *ogbg-molpcba* that has only categorical values).
The first baseline is a simple GAUSSIAN distribution, which computes its sufficient statistics (mean and standard deviation) from the training set and then computes the NLL on the dataset. The second baseline is a structure-agnostic Gaussian Mixture Model (GMM), which does not rely on the structure when performing imputation but can model arbitrarily complex distributions; we recall that our model behaves like a GMM when the number of layers is set to 1. The set of hyper-parameters configurations tried for each baseline is reported in Table 5.

| Model
Eval. Process | GIN
Sup. | GAE$_{+DS}$
Unsup.$\to$ Sup. | DGI$_{+DS}$
Unsup.$\to$ Sup. | GSPN$_{U+DS}$
Unsup.$\to$ Sup. |
|---|---|---|---|---|
| benzene | $41.4 \pm 45.6$ | $\mathbf{2.00} \pm 0.1$ | $6.22 \pm 1.2$ | $\underline{2.24} \pm 0.5$ |
| ethanol | $4.00 \pm 1.0$ | $7.79 \pm 7.8$ | $5.35 \pm 3.5$ | $\mathbf{3.36} \pm 0.0$ |
| malonaldehyde | $8.00 \pm 5.3$ | $5.49 \pm 1.3$ | $\mathbf{4.31} \pm 1.8$ | $\mathbf{4.31} \pm 1.4$ |
| naphthalene | $34.9 \pm 19.4$ | $4.45 \pm 0.3$ | $6.34 \pm 2.3$ | $\mathbf{4.44} \pm 0.1$ |
| salicylic acid | $36.7 \pm 13.4$ | $233.3 \pm 27.1$ | $17.0 \pm 7.3$ | $\mathbf{4.90} \pm 0.5$ |
| toluene | $29.6 \pm 18.0$ | $\mathbf{4.83} \pm 0.3$ | $9.18 \pm 0.3$ | $\underline{4.87} \pm 1.4$ |
| uracil | $19.7 \pm 14.1$ | $387.7 \pm 13.7$ | $409.7 \pm 1.5$ | $\mathbf{3.93} \pm 0.0$ |
| ogbg-molpcba ($\uparrow$) | $\mathbf{4.12} \pm 0.2$ | $3.36 \pm 0.3$ | $3.00 \pm 0.5$ | $\underline{4.00} \pm 0.5$ |

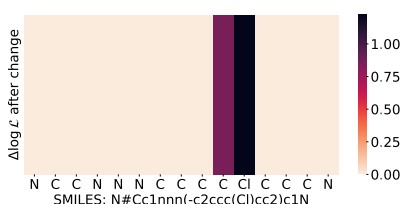

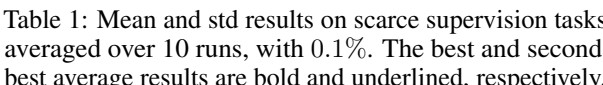

SMILES: N#Cc1nnn(-c2ccc(Cl)cc2)c1N

Table 1: Mean and std results on scarce supervision tasks averaged over 10 runs, with $0.1\%$. The best and second-best average results are bold and underlined, respectively.

Figure 3: Relative change in vertices pseudo log-likelihood when replacing *Cl* in the SMILES with an *O*.

**Graph Classification**  Among the graph classification tasks of Errica et al. (2020), we restrict our evaluation to NCI1 (Wale et al., 2008), REDDIT-BINARY, REDDIT-MULTI-5K, and COLLAB (Yanardag & Vishwanathan, 2015) datasets, for which leveraging the structure seems to help improving the performances. The empirical setup and data splits are the same as Errica et al. (2020) and Castellana et al. (2022), from which we report previous results. The baselines we compare against are a structure-agnostic baseline (BASELINE), DGCNN (Zhang et al., 2018a), DiffPool (Ying et al., 2018), ECC (Simonovsky & Komodakis, 2017), GIN (Xu et al., 2019), GraphSAGE (Hamilton et al., 2017a), CGMM, E-CGMM (Atzeni et al., 2021), and iCGMM (Castellana et al., 2022). CGMM can be seen as an incrementally trained version of GSPN that cannot use probabilistic shortcut connections. Table 6 shows the set of hyper-parameters tried for GSPN, and Table 9 shows that the time to compute a forward and backward pass of GSPN is comparable to GIN.

## 6  RESULTS

We present quantitative and qualitative results following the structure of Section 5. For all experiments we used a server with 32 cores, 128 GBs of RAM, and 8 Tesla-V100 GPUs with 32 GBs of memory.

**Scarce Supervision**  In Table 1, we report the performance of GSPN$_U$ combined with DeepSets (DS) in the scarce supervision scenario. We make the following observations. First, a fully supervised model like GIN is rarely able to perform competitively when trained only on a small amount of labeled data. This is not true for *ogbg-molpcba*, but the AP scores are also very low, possibly because of the high class imbalance in the dataset. Secondly, on the first seven chemical tasks the unsupervised embeddings learned by GAE and DGI sometimes prevent DS from converging to a stable solution. This is the case for *ethanol*, *salicylic acid*, and *uracil*, where the corresponding MAE is higher than those of the GIN model. GSPN is competitive with and often outperforms the alternative approaches, both in terms of mean and standard deviation, and it ranks first or second place across all tasks. These results suggest that modeling the distribution of vertex attributes conditioned on the graph is a good inductive bias for learning meaningful vertex representations in the chemical domain and that exploiting a large amount of unsupervised graph data can be functional to solving downstream tasks in a scarce supervision scenario.

Contrarily to the other baselines, GSPN can answer probabilistic queries about the graph as depicted in Figure 3. Here we replace the Chlorine (*Cl*) atom in the SMILES of an *ogbg-molpcba* sample with an Oxygen (*O*) and ask the model to compute again the pseudo log-likelihood of each vertex (Equation 1). In this case, it makes intuitive sense that the relative likelihood increases because *Cl* is deactivating, so it is more unlikely to observe it attached to the all-carbon atom group. Being able to query the probabilistic model subject to changes in the input naturally confers a degree of interpretability and trustworthiness to the model. For the interested reader, we provide more visualizations for randomly picked samples in Section A.9.

**Modeling the Data Distribution under Partial Evidence**  We demonstrate that is beneficial to model structural dependencies with GSPN to handle missing values in Table 2. The first observation is that modeling multimodality using a GMM already improves the NLL significantly over the unimodal GAUSSIAN baseline; though this may not seem surprising at first, we stress that this result

| | benzene | ethanol | malonaldehyde | naphthalene | salicylic acid | toluene | uracil | S | M |
|---|---|---|---|---|---|---|---|---|---|
| GAUSSIAN | $9.96 \pm 0.0$ | $5.72 \pm 0.0$ | $5.88 \pm 0.0$ | $6.48 \pm 0.0$ | $6.91 \pm 0.0$ | $6.00 \pm 0.0$ | $7.54 \pm 0.0$ | ✗ | ✗ |
| GMM | $4.31 \pm 0.01$ | $3.81 \pm 0.01$ | $3.93 \pm 0.01$ | $3.31 \pm 0.10$ | $3.21 \pm 0.13$ | $3.59 \pm 0.06$ | $\mathbf{3.11} \pm 0.5$ | ✗ | ✓ |
| GSPN$_U$ | $\mathbf{4.17} \pm 0.02$ | $\mathbf{3.77} \pm 0.03$ | $\mathbf{3.88} \pm 0.01$ | $\mathbf{3.18} \pm 0.07$ | $\mathbf{3.06} \pm 0.21$ | $\mathbf{3.35} \pm 0.05$ | $3.17 \pm 0.17$ | ✓ | ✓ |

Table 2: We report the mean and std NLL under the missing data scenario, averaged over 3 runs. Symbols S and M stand for "uses structure" and "captures multimodality", respectively.

| | NCI1 | REDDIT-B | REDDIT-5K | COLLAB |
|---|---|---|---|---|
| Baseline | $69.8 \pm 2.2$ | $82.2 \pm 3.0$ | $52.2 \pm 1.5$ | $70.2 \pm 1.5$ |
| DGCNN | $76.4 \pm 1.7$ | $87.8 \pm 2.5$ | $49.2 \pm 1.2$ | $71.2 \pm 1.9$ |
| DIFFPOOL | $76.9 \pm 1.9$ | $89.1 \pm 1.6$ | $53.8 \pm 1.4$ | $68.9 \pm 2.0$ |
| ECC | $76.2 \pm 1.4$ | - | - | - |
| GIN | $\mathbf{80.0} \pm 1.4$ | $89.9 \pm 1.9$ | $\mathbf{56.1} \pm 1.7$ | $75.6 \pm 2.3$ |
| GRAPHSAGE | $76.0 \pm 1.8$ | $84.3 \pm 1.9$ | $50.0 \pm 1.3$ | $73.9 \pm 1.7$ |
| CGMM$_{U+DS}$ | $76.2 \pm 2.0$ | $88.1 \pm 1.9$ | $52.4 \pm 2.2$ | $77.3 \pm 2.2$ |
| E-CGMM$_{U+DS}$ | $\underline{78.5} \pm 1.7$ | $89.5 \pm 1.3$ | $53.7 \pm 1.0$ | $77.5 \pm 2.3$ |
| ICGMM$_{U+DS}$ | $77.6 \pm 1.5$ | $\mathbf{91.6} \pm 2.1$ | $\underline{55.6} \pm 1.7$ | $\mathbf{78.9} \pm 1.7$ |
| GSPN$_{U+DS}$ | $^\dagger 76.6 \pm 1.9$ | $^\dagger \underline{90.5} \pm 1.1$ | $^\dagger 55.3 \pm 2.0$ | $^\dagger \underline{78.1} \pm 2.5$ |
| GSPN$_S$ | $^\dagger 77.6 \pm 3.0$ | $^\dagger 89.7 \pm 2.3$ | $^\dagger 54.2 \pm 2.1$ | $74.1 \pm 2.5$ |

Table 3: Mean and std results on graph classification datasets. The best and second-best performances are bold and underlined, respectively. A † means that GSPN improves over CGMM.

is a good proxy to measure the usefulness of the selected datasets in terms of different possible values, and it establishes an upper bound for the NLL.

Across almost all datasets GSPN improves the NLL score, and we attribute this positive result to its ability to simultaneously capture both the structural dependencies and the multimodality of the data distribution. The only dataset where the GMM has a better NLL is uracil, and we argue that this might be due to the independence of the node attributes from the surrounding structural information. These empirical results seem to agree with the considerations of Sections 4.2 and shed more light on a research direction worthy of further investigations.

**Graph Classification** The last quantitative analysis concerns graph classification, whose results are reported in Table 3. As we can see, not only is GSPN competitive against a consistent set of neural and probabilistic DGNs, but it also improves almost always w.r.t. $CGMM$, which can be seen as the layer-wise counterpart of our model. In addition, the model ranks second on two out of five tasks, though there is no clear winner among all models and the average performances are not statistically significant due to high variance. We also observe that here GSPN$_S$ does not bring significant performance gains compared to GSPN$_{U+DS}$. As mentioned in Section 4.4, we attribute this to the limited theoretical expressiveness of the global aggregation function. We also carry out a qualitative analysis on the impact of the number of layers, $C$ and $C_G$ for GSPN$_S$ (with global sum pooling) on the different datasets, to shed light on the benefits of different hyper-parameters. The results show that on NCI1 and COLLAB the performance improvement is consistent as we add more layers, whereas the opposite is true for the other two datasets where the performances decrease after five layers. Therefore, the selection of the best number of layers to use remains a non-trivial and dataset-specific challenge. In the interest of space, we report a visualization of these results in Appendix A.8.

## 7 CONCLUSIONS

We have proposed Graph-Induced Sum-Product Networks, a deep and fully probabilistic class of models that can tractably answer probabilistic queries on graph-structured data. GSPN is a composition of locally valid SPNs that mirrors the message-passing mechanism of DGNs, and it works as an unsupervised or supervised model depending on the problem at hand. We empirically demonstrated its efficacy across a diverse set of tasks, such as scarce supervision, modeling the data distribution under missing values, and graph prediction. Its probabilistic nature allows it to answer counterfactual queries on the graph, something that most DGNs cannot do. We hope our contribution will further bridge the gap between probabilistic and neural models for graph-structured data.

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

# A APPENDIX

We complement the discussion with theoretical and practical considerations that facilitate understanding and reproducibility.

## A.1 HIGH-LEVEL DEFINITIONS OF PROBABILISTIC CIRCUITS

We introduce basic concepts of PCs to ease the understanding of readers that are less familiar with this topic.

A probabilistic circuit (Choi et al., 2020) over a set of *r.v.*s $X = (A_1, \ldots, A_d)$ is uniquely determined by its circuit structure and computes a (possibly unnormalized) distribution $P(X)$. The circuit structure has the form of a rooted DAG, which comprises a set of computational units. In particular, *input* units are those for which the set of incoming edges is empty (i.e., no children), whereas the *output* unit has no outgoing edges.

The *scope* of a PC is a function that associates each unit of the circuit with a subset of $X$. For each non-input unit, the scope of the unit is the union of the scope of the children. It follows that the scope of the root is $X$.

Each *input* (or *distribution*) unit encodes a parametric non-negative function, e.g., a Gaussian or Categorical distribution. A *product* unit represents the joint, fully factorized distribution between the distributions encoded by its children. Finally, a *sum* unit defines a weighted sum of the children's distributions, i.e., a mixture model when the children encode proper distributions. The set of parameters of a probabilistic circuit is given by the union of the parameters of all input and sum units in the circuit.

There are different kinds of probabilistic queries that a PC can answer. The first is the *complete evidence* query, corresponding to the computation of $P(X = x)$. A single feedforward pass from the input units to the output unit is sufficient to compute this query. Another important class of queries is that of *marginals*, where we assume that not all *r.v.*s are fully observed, e.g., missing values. Given partial evidence $E \subset X$ and the unobserved variables $Z = X \setminus E$, a marginal query is defined as $P(E = e) = \int p(e, z) dz$. Finally, we mention the *conditional* query, sharing the same complexity as the marginal, where we compute the conditional probability $P(Q = q \mid E = e)$ of a subset of *r.v.*s $Q \subset X$ conditioned on partial evidence $E$ and $Z = X \setminus (E \cup Q)$.

To be able to *tractably* compute the above queries, one usually wants the PC to have specific structural properties that guarantee a linear time complexity for marginal and conditional queries. A product unit is said to be *decomposable* when the scopes of its children are disjoint, and a PC is decomposable if all its product units are decomposable. Instead, a sum unit is *smooth* if all its children have identical scopes, and a PC is smooth (or *complete* (Poon & Domingos, 2011)) if all its sum units are smooth. For instance, the NB model considered in our work is implemented as a smooth and decomposable PC.

A PC that is smooth and decomposable can tractably compute marginals and conditional queries (one can prove that these are necessary and sufficient conditions for tractable computations of these queries), and we call such SPNs *valid*. A generalization of decomposability, namely *consistency*, is necessary to tractably compute maximum a posteriori queries of the form $\arg\max_q P(Q = q, E = e)$.

PCs have an interpretation in terms of latent variable models (Peharz et al., 2016), and in particular it is possible to augment PCs with specialized input units that mirror the latent variables of the associated graphical model. However, it is not immediate at all to see that *valid* SPNs allow a tractable computation of the posterior probabilities for the sum units, and we indeed refer the reader to works in the literature that formally prove it. As stated in (Poon & Domingos, 2011; Peharz et al., 2016), inference in unconstrained SPNs is generally intractable, but when an SPN is valid efficient inference is possible. Concretely, one can get all the required statistics for computing the posterior probabilities $h_j$ of any sum unit $j$ in a single backpropagation pass across the SPN (Equation 22 of Peharz et al. (2016)). The posterior computation involves the use of tractable quantities and hence stays tractable. Of course, the computational costs depend on the size of the SPN and might be impractical if the SPN is too large, but we still consider it tractable in terms of asymptotic time complexity.

## A.2 THE SWITCHING PARENT DECOMPOSITION

The decomposition used in Equation 4, known as "Switching Parent" (SP) in the literature (Bacciu et al., 2010), was formally introduced in (Saul & Jordan, 1999) in the context of mixed memory Markov models. We report the original formulation below. Let $i_t \in \{1, \dots, n\}$ denote a discrete random variable that can take on $n$ possible values. Then we write the SP decomposition as

$$P(i_t \mid i_{t-1}, \dots, i_{t-k}) = \sum_{\mu=1}^{n} P(\mu) P^{\mu}(i_t \mid i_{t-\mu}). \tag{6}$$

The connection to Equation 4 emerges by observing the following correspondences: $P(\mu)$ is treated as a constant $\frac{1}{|ch_n|}$, $P^{\mu}(i_t = i \mid i_{t-\mu} = j)$ implements the *transition* conditional probability table, which we model in a "soft" version (see also (Bacciu et al., 2020a)) as $\boldsymbol{\theta}^{\ell} \boldsymbol{h}_u^{\ell-1}$, and the conditional variables on the left-hand-side of the equation intuitively correspond to the information $\boldsymbol{h}_{ch_n}^{\ell-1}$ we use to parametrize the prior $P_{\boldsymbol{\pi}^{\ell}}(Q_v^{\ell})$. Moreover, we assume full stationarity and ignore the position of the child in the parametrization, meaning we use the same transition weights $\boldsymbol{\theta}^{\ell}$ for all neighbors; this is crucial since there is usually no consistent ordering of the vertices across different graphs, and consequently between the children in the SPN hierarchy.

## A.3 PROOF THAT EQUATION 4 IS A VALID PARAMETRIZATION

To show that the computation $\boldsymbol{\pi}_{n'}^{\ell+1}$ outputs a valid parametrization for the categorical distribution $P_{\boldsymbol{\pi}_{n'}^{\ell+1}}(Q_{n'}^{\ell+1})$, it is sufficient to show that the parameters sum to 1:

$$
\begin{aligned}
\sum_{i=1}^{C} \boldsymbol{\pi}_{n'}^{\ell+1}(i) = \sum_{i=1}^{C} f_{\boldsymbol{\theta}^{\ell+1}}(\boldsymbol{h}_1^{\ell}, \dots, \boldsymbol{h}_{|ch_{n'}|}^{\ell})_i &= \sum_{i=1}^{C} \frac{1}{|ch_{n'}|} \sum_{n \in ch_{n'}} \sum_{k=1}^{C} \boldsymbol{\theta}_{ki}^{\ell+1} \boldsymbol{h}_n^{\ell}(k) \\
&= \frac{1}{|ch_{n'}|} \sum_{n \in ch_{n'}} \sum_{k=1}^{C} \sum_{i=1}^{C} \boldsymbol{\theta}_{ki}^{\ell+1} \boldsymbol{h}_n^{\ell}(k) \\
&= \frac{1}{|ch_{n'}|} \sum_{n \in ch_{n'}} \sum_{k=1}^{C} \boldsymbol{h}_n^{\ell}(k) = \frac{1}{|ch_{n'}|} \sum_{n \in ch_{n'}} 1 = 1, \quad (7)
\end{aligned}
$$

where we used the fact that the rows of $\boldsymbol{\theta}^{\ell}$ and the posterior weights $\boldsymbol{h}_u^{\ell-1}$ are normalized.

A.4  DEALING WITH MORE GENERAL SPN TEMPLATES

In Section 4 we have introduced the general framework of GSPN for arbitrary SPNs, but the explicit implementation of $\boldsymbol{\pi}_{n,j}^{\ell} = f_{\boldsymbol{\theta}_j^{\ell}}(\boldsymbol{h}_{n_1,j}^{\ell-1}, \ldots, \boldsymbol{h}_{n_T,j}^{\ell-1})$ and the computation of the posterior probabilities $\boldsymbol{h}_{n_i,j}^{\ell-1}$ has only been shown for the Naïve Bayes model (Section 4.1) with a *single* sum unit $j$ in its corresponding SPN template. Despite that the computation of the posterior depends on the specific template used, we can still provide guidelines on how to use GSPN in the general case.

Consider any *valid* SPN template with $S$ sum units, and *w.l.o.g.* we can assume that all sum units have $C$ different weights, i.e., each sum unit implements a mixture of $C$ distributions. Given a computational tree, we consider an internal node $n$ with $T$ children $n_1, \ldots, n_T$, and we recall that all SPNs associated with the nodes of the tree share the same template (although a different parametrization). Therefore, there is a one-to-one correspondence between unit $j$ of node $n_i$, $\forall i \in \{1, \ldots, T\}$, at level $\ell - 1$ and unit $j$ of its parent $n$ at level $\ell$, meaning that the parametrization $\boldsymbol{\pi}_{n,j}^{\ell} \in \mathbb{R}^C$ can be computed using a permutation invariant function such as

$$\boldsymbol{\pi}_{n,j}^{\ell} = f_{\boldsymbol{\theta}_j^{\ell}}(\boldsymbol{h}_{n_1,j}^{\ell-1}, \ldots, \boldsymbol{h}_{n_T,j}^{\ell-1}) = \frac{1}{T} \sum_{i=1}^{T} \boldsymbol{\theta}_j^{\ell} \boldsymbol{h}_{n_i,j}^{\ell-1} \quad \forall j \in \{1, \ldots, S\} \tag{8}$$

that acts similarly to the neighborhood aggregation function of DGNs.

All that remains is to describe how we compute the posterior probabilities $\boldsymbol{h}_{n,j}^{\ell} \in \mathbb{R}^C$ for all sum units $j \in \{1, \ldots, S\}$ of a generic node $n$ at level $\ell$ (now that the reader is familiar with the notation, we can abstract from the superscript $\ell$ since it can be determined from $n$). As discussed in Section A.1, SPNs have an interpretation in terms of latent variable models, so we can think of a sum unit $j$ as a latent random variable and we can augment the SPN to make this connection explicit (Peharz et al., 2016). Whenever the SPN is valid, the computation of the posterior probabilities of all sum units is *tractable* and it requires *just one* backpropagation pass across the augmented SPN (see Equation 22 of Peharz et al. (2016)). This makes it possible to tractably compute the posterior probabilities of the sum units that will be used to parametrize the corresponding sum units of the parent node in the computational tree. Below we provide a pseudocode summarizing the inference process for a generic GSPN, but we remind the reader that the message passing procedure is equivalent to that of DGNs.

---

**Algorithm 1** GSPN Inference on a Single Computational Tree

---

      **Input:** Computational tree with $T$ nodes and height $L$, valid SPN template with $S$ sum units and $C$ weights for each sum unit.
      **Output:** Set of posterior probabilities $\{\boldsymbol{h}_{n,j} \ \forall n \in \{1, \ldots, T\}, \forall j \in \{1, \ldots, S\}\}$
1: **for** $\ell = 0, \ldots, L$ **do**
2:     **for** all nodes $n$ at height $\ell$ **do**
3:         **for** all sum units $j$ of SPN of node $n$ **do**:
4:             **if** $\ell > 0$ **then**:
5:                 Compute $\boldsymbol{\pi}_{n,j}$ (Eq. 8)           ▷ message passing, parallelized
6:             **else**
7:                 Use learned $\boldsymbol{\pi}_{n,j}$            ▷ leaf node, no children
8:             **end if**
9:             Compute and collect $\boldsymbol{h}_{n,j}$ (Equation 22 (Peharz et al., 2016)) ▷ computation reused across sum units
10:         **end for**
11:     **end for**
12: **end for**
13: **return** $\{\boldsymbol{h}_{n,j} \ \forall n \in \{1, \ldots, T\}, \forall j \in \{1, \ldots, S\}\}$

---

On a separate note, when it comes to handling missing data, it is sufficient to marginalize out the missing evidence by substituting a 1 in place of the missing input units. This allows us to compute any marginal or conditional query (including the computation of the posterior probabilities) in the presence of partial evidence.

## A.5 PROBABILISTIC SHORTCUT CONNECTIONS

In $\text{GSPN}_U$, we also propose probabilistic shortcut connections to set the shared parameters $\boldsymbol{\omega}^L$ as a convex combination of those at height $\{\boldsymbol{\omega}^0, \ldots, \boldsymbol{\omega}^{L-1}\}$. For instance, for a continuous *r.v.* $\tilde{\boldsymbol{X}}_n^L$ of the root node $n$, a modeling choice would be to take the mean of the $L-1$ Gaussians implementing the distributions, leveraging known statistical properties to obtain

$$P_{\boldsymbol{\omega}^L}(\tilde{\boldsymbol{X}}_n^L \mid Q_n^L = i) \stackrel{\text{def}}{=} \mathcal{N}\left(\cdot\,; \sum_{\ell=1}^{L-1} \frac{\mu_i^\ell}{L-1}, \sum_{\ell=1}^{L-1} \frac{(\sigma_i^\ell)^2}{(L-1)^2}\right), \quad \boldsymbol{\omega}^\ell = (\mu_1^\ell, \sigma_1^\ell, \ldots, \mu_C^\ell, \sigma_C^\ell). \quad (9)$$

Instead, when the *r.v.* is categorical, we consider a newly parametrized Categorical distribution:

$$P_{\boldsymbol{\omega}^L}(\tilde{\boldsymbol{X}}_n^L \mid Q_n^L = i) \stackrel{\text{def}}{=} Cat\left(\cdot\,; \sum_{\ell=1}^{L-1} \frac{\boldsymbol{\omega}_i^\ell}{L-1}\right), \boldsymbol{\omega}_i^\ell \in C\text{-1-probability simplex } \forall i \in [1, C]. \quad (10)$$

Akin to residual connections in neural networks (Srivastava et al., 2015; He et al., 2016), these shortcut connections mitigate the vanishing gradient and the degradation (accuracy saturation) problem, that is, the problem of more layers leading to higher training error. In the experiments, we treat the choice of using residual connections as a hyper-parameter and postpone more complex design choices, such as *weighted* residual connections, to future work.

## A.6 HYPER-PARAMETERS TRIED DURING MODEL SELECTION

The following tables report the hyper-parameters tried during model selection.

| | **Embedding Construction** | | | | | | |
| --- | --- | --- | --- | --- | --- | --- | --- |
| | $C$ / latent dim | # layers | learning rate | batch size | # epochs | ES patience | avg emission params across layers |
| GAE | 32, 128, 256 | 2,3,5 | 0,1, 0.01 | 1024 | 100 | 50 | |
| DGI | 32, 128, 256 | 2,3,5 | 0,1, 0.01 | 1024 | 100 | 50 | |
| GSPN | 5,10,20,40 | 5,10,20 | 0.1 | 1024 | 100 | 50 | true, false |
| | **Graph Predictor** | | | | | | |
| | | | | | | global pooling | w. decay (MLP) dropout (GIN) |
| MLP | 8,16,32,64 | 1 | 0.01 | 1024 | 1000 | 500 | sum, mean | 0, 0.0001 |
| GIN | 32,256,512 | 2,5 | 0.01, 0.0001 | 8,32,128 | 1000 | 500 | sum, mean | 0., 0.5 |

Table 4: Scarce supervision experiments. We found that too large batch sizes caused great instability in GIN's training, so we tried different, smaller options. DGI and GAE used the Atom Embedder of size 100 provided by OGBG, whereas GSPN deals with categorical attributes through a Categorical distribution. The range of hyper-parameters tried for GAE and DGI follows previous works.

| | $C$ | # layers | learning rate | batch size | # epochs | ES patience | avg emission params across layers |
| --- | --- | --- | --- | --- | --- | --- | --- |
| GAUSSIAN | - | - | - | - | - | - | - |
| GMM | 5,15,20,40 | 1 | 0,1, 0.01 | 32 | 200 | 50 | - |
| GSPN | 5,15,20,40 | 2 | 0,1, 0.01 | 32 | 200 | 50 | false |

Table 5: Missing data experiments. Gaussian mixtures of distributions are initialized with k-means on the first batch of the training set. The maximum variance at initialization is set to 10.

| | $C$ / latent dim | # layers | learning rate | batch size | # epochs | ES patience | avg emission params across layers |
|---|---|---|---|---|---|---|---|
| | | | **Embedding Construction** | | | | |
| $\text{GSPN}_U$ | 5,10,20 | 5, 10, 15, 20 | 0,1 | 32 | 500 | 50 | true, false |
| | | | **Graph Predictor** | | | | |
| | | | | | | | global pooling |
| MLP | 8, 16, 32, 128 | 1 | 0.001 | 32 | 1000 | 200 | sum, mean |
| $\text{GSPN}_S$ | $C$=5,10,20 $C_g$=32,128 | 1 | 0.001 | 32 | 1000 | 200 | sum, mean |

Table 6: Graph classification experiments. The emission distribution is categorical for NCI1 and univariate Gaussian otherwise. Gaussian mixtures of distributions to be learned (social datasets) are initialized using k-means on the training set.

## A.7 DATASETS STATISTICS

Below we report the set of datasets used in our work together with their characteristics.

| | # graphs | # vertices | # edges | # vertex attributes | task | metric |
|---|---|---|---|---|---|---|
| benzene | 527984 | 12.00 | 64.94 | 1 categorical + 6 cont. | Regression(1) | MAE/NLL |
| ethanol | 455093 | 9.00 | 36.00 | 3 (1 cat.) + 6 cont. | Regression(1) | MAE/NLL |
| naphthalene | 226256 | 18.00 | 127.37 | 3 (1 cat.) + 6 cont. | Regression(1) | MAE/NLL |
| salicylic acid | 220232 | 16.00 | 104.13 | 3 (1 cat.) + 6 cont. | Regression(1) | MAE/NLL |
| toluene | 342791 | 15.00 | 96.15 | 3 (1 cat.) + 6 cont. | Regression(1) | MAE/NLL |
| malonaldehyde | 893238 | 9.00 | 36.00 | 3 (1 cat.) + 6 cont. | Regression(1) | MAE/NLL |
| uracil | 133770 | 12.00 | 64.44 | 4 (1 cat.) + 6 cont. | Regression(1) | MAE/NLL |
| ogbg-molpcba | 437929 | 26.0 | 28.1 | 83 (9 cat.) | Multi-Label Classification(128) | AP |
| NCI1 | 4110 | 29.87 | 32.30 | 37 (1 cat.) | Classification(2) | ACC |
| REDDIT-B | 2000 | 429.63 | 497.75 | 1 cont. | Classification(2) | ACC |
| REDDIT-5K | 5000 | 508.52 | 594.87 | 1 cont. | Classification(5) | ACC |
| COLLAB | 5000 | 74.49 | 2457.78 | 1 cont. | Classification(3) | ACC |

Table 7: Dataset statistics.

## A.8 GRAPH CLASSIFICATION EXPERIMENTS: ABLATION AND HYPER-PARAMETER STUDIES

**Ablation Study** The following table shows the performance of $\text{GSPN}_{U+DS}$ when removing the use shortcut connections from the hyper-parameter space. The results show that using shortcut connections consistently leads to better mean classification accuracy on these tasks.

| | **NCI1** | **REDDIT-B** | **REDDIT-5K** | **COLLAB** |
|---|---|---|---|---|
| $\text{GSPN}_{U+DS}$ (no shortcut) | $76.5 \pm 1.9$ | $88.9 \pm 3.9$ | $52.3 \pm 5.2$ | $75.7 \pm 2.6$ |
| $\text{GSPN}_{U+DS}$ | $76.6 \pm 1.9$ | $90.5 \pm 1.1$ | $55.3 \pm 2.0$ | $78.1 \pm 2.5$ |

Table 8: Impact of shortcut connections on the creation of unsupervised embeddings for graph classification.

**Impact of Hyper-parameters** Figure 4 shows how the validation performance of $\text{GSPN}_S$ changes for specific hyper-parameters $C$ and $C_G$ as we add more layers. Please refer to the main text for a discussion.

## A.9 SCARCE SUPERVISION EXPERIMENTS: ADDITIONAL VISUALIZATIONS

Figure 5 provides a few, randomly picked examples of molecules from the *ogbg-molpcba* dataset, modified to show the relative change in pseudo log-likelihood of some of the vertices according to GSPN.

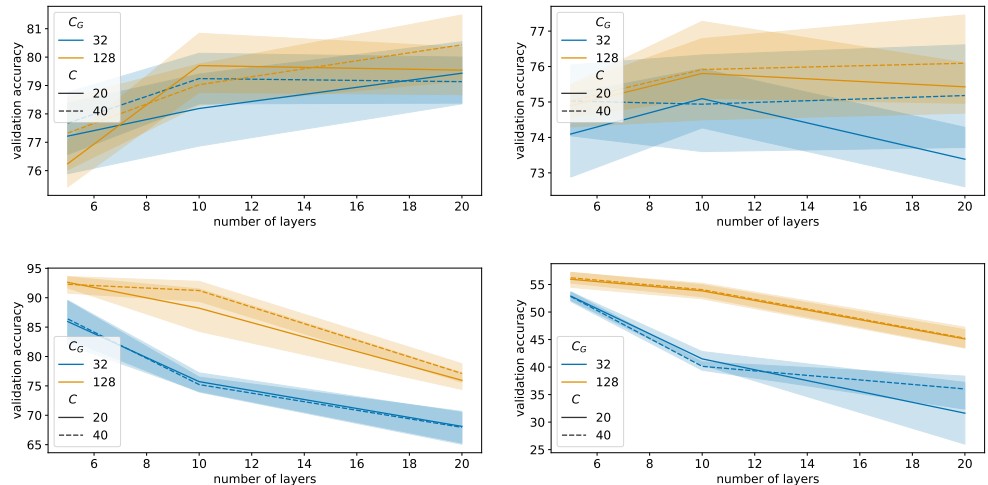

Figure 4: Impact of GSPN$_S$ layers, $C$ and $C_G$ on NCI1 (top left), COLLAB (top right), REDDIT-BINARY (bottom left), and REDDIT-5K (bottom right) performances, averaged across all configurations in the 10 outer folds.

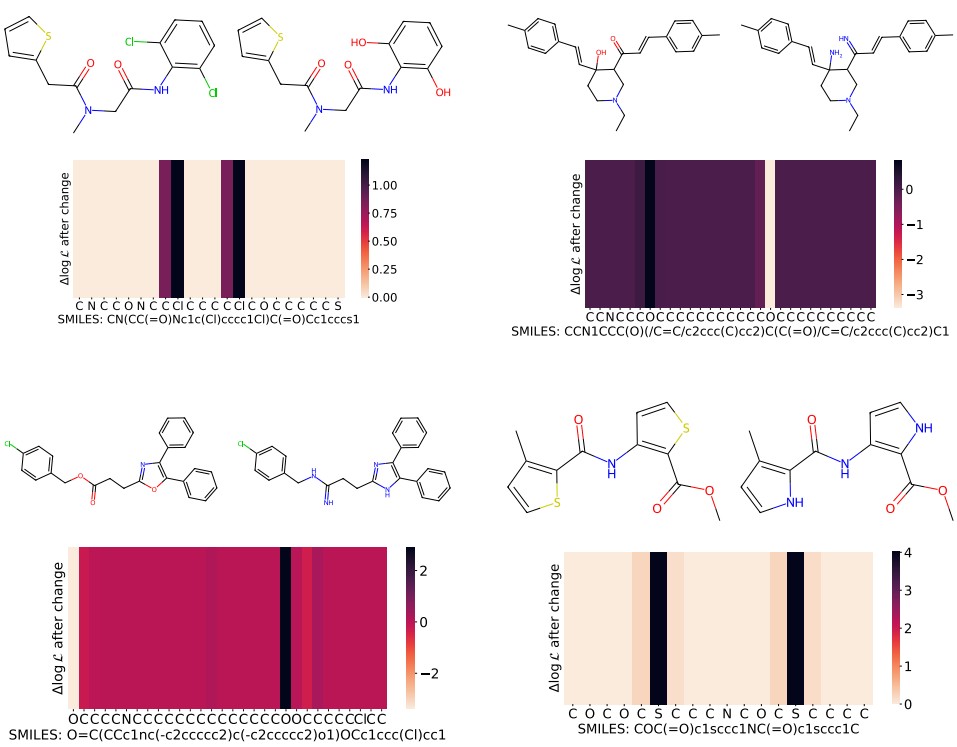

Figure 5: Additional visualizations of randomly picked molecules from *ogbg-molpcba* for a specific GSPN configuration. The heatmap shows pseudo log-likelihood variations in the vertices when operating specific atomic changes (from left to right). Nearby atoms are affected by the change as well, and this allows us to understand which groups are more common, or make more sense, than others.

## A.10 TIME COMPARISON

Table 9 shows the time comparison of forward and backward passes on a batch of size 32 between GSPN and GIN. We used 10 layers for both architectures and adapted the hidden dimensions to obtain a comparable number of parameters. Despite the GIN's implementation being more sample efficient than GSPN, the table confirms our claims on the asymptotic complexity of GSPN.

| | # parameters | | forward time (ms) | | backward time (ms) | |
|---|---|---|---|---|---|---|
| | GSPN | GIN | GSPN | GIN | GSPN | GIN |
| NCI1 | 36876 | 36765 | 30 | 12 | 17 | 14 |
| REDDIT-B | 34940 | 35289 | 35 | 14 | 24 | 15 |
| REDDIT-5K | 34940 | 35289 | 35 | 14 | 22 | 15 |
| COLLAB | 34940 | 35289 | 36 | 14 | 21 | 15 |

Table 9: Time comparison between GSPN and GIN on graph classification tasks.

