# OpenReview forum: "Tractable Probabilistic Graph Representation Learning with Graph-Induced Sum-Product Networks"
_ICLR.cc/2024/Conference — ICLR 2024 poster_

### Official Review · Reviewer_GF8k · 2023-10-26

**Soundness:** 3 good
**Presentation:** 2 fair
**Contribution:** 3 good
**Rating:** 6
**Confidence:** 2

**Summary:**

The paper suggests utilizing tractable probabilistic models (TPMs) in graph representation learning. More precisely, the proposed graph-induced sum-product networks (GSPNs) are a class of hierarchies of sum-product networks (SPNs) that are capable of answering many types of probabilistic queries in a polynomial time in the size of the model. Further, the authors show that GSPNs are capable of dealing with missing data in the graph. The theoretical results are complemented by empirical experiments, where the authors show that are GSPNs are competitive on the tasks of scarce supervision, modeling data distributions with missing values, and graph classification.

**Strengths:**

I was unable to grasp every detail of the paper due to my limited knowledge of some of the topics, so please take my review with a grain of salt.

The main contribution of the paper is introducing the GSPN framework on the active area of graph representation learning with the following important properties:
- Efficiently computable probabilistic queries
- The ability to deal with missing data
- The ability to approximate the joint distribution over the random variables, where the graph can have arbitrary structure

Further, the empirical experiments demonstrate that the proposed class of structures is not only theoretically interesting but also in practice.

**Weaknesses:**

The paper could be more polished:
- As per formatting instructions, the citations should be in parenthesis when they are not part of a sentence.
- "The neighborhood of a vertex $v$ is the set $N_v = \lbrace u \in V | (u, v) \in E\rbrace$ of incoming edges": the neighborhood is not a set of edges but vertices.
- Section 6 starts with a lorem ipsum paragraph

**Questions:**

If I understood correctly, a small height L of the trees for graphs with a large diameter (consider, e.g., an n-cycle) would result in the trees containing only few of the vertices of the graph. On the other hand, a large L leads to an exponential blowup in the size of the trees, which is computationally infeasible. Is having the trees contain only few of the vertices of the graph detrimental, and if yes, then how harmful is it?

---

> ### Author Response · Authors · 2023-11-15
> **Response to Reviewer GF8k**
>
> We thank the reviewer for highlighting the strengths of GSPNs, and for giving us actionable feedback to improve the manuscript. Below, we answer each comment or question in order.
>
> ### Fixes:
> - We have adjusted the citations’ style as suggested.
> - We have removed the incorrect statement about the definition of the neighborhood.
> - We have removed the lorem ipsum paragraph. Apologies for this.
>
> ### Questions:
>
> **Q:** If I understood correctly, a small height L of the trees for graphs with a large diameter (consider, e.g., an n-cycle) would result in the trees containing only few of the vertices of the graph. On the other hand, a large L leads to an exponential blowup in the size of the trees, which is computationally infeasible. Is having the trees contain only few of the vertices of the graph detrimental, and if yes, then how harmful is it?
>
> **A:** The understanding of the reviewer is correct, $L$ controls the “receptive field” of each vertex akin to standard message-passing neural networks of depth $L$ (GSPN is in fact implemented as a message-passing architecture of depth $L$). Due to space reasons, we had to defer the impact of the depth for some tasks to Appendix A.8. In general, we observed that $L$ is a crucial hyper-parameter of the network that needs to be appropriately chosen for a given task. A small height $L$ might suffice in some cases (for instance the two REDDIT datasets) and not in others (for instance in NCI1 and COLLAB), where more information needs to be propagated across the graph.
>
> --
>
> We hope our answers have clarified the doubts of the reviewer. Thank you again for helping us improve the manuscript, and we hope that with the current modifications the reviewer will consider raising the score.

---

> > ### Comment · Reviewer_GF8k · 2023-11-20
> > **Response to the Authors**
> >
> > I thank the authors for their response and clarifying the interpretation of the hyperparameter L. However, due to my limited expertise on some of the topics, I will refrain from changing my score to avoid adding noise to the reviewing process. Still, I wish to emphasize that I appreciate the authors revising their paper based on the feedback.

---

### Official Review · Reviewer_KgLo · 2023-11-03

**Soundness:** 3 good
**Presentation:** 4 excellent
**Contribution:** 2 fair
**Rating:** 6
**Confidence:** 3

**Summary:**

The paper suggests a framework for using Sum-Product Networks (SPNs) as Deep Graph Networks (DGNs). That is, the framework establish methods for representing a computation graph, such as the ones used in neural network architectures, as an SPN. The practical motivations for the work comes from DGNs, in general: (i) having overconfident predictions due to lack of uncertainty consideration, (ii) ad-hoc imputation method of missing data due to lack of probabilistic queries.
The authors suggest solving these issues by representing a DGN as a hierarchy of interconnected SPN and, therefore, being capable of answering probabilistic queries in a tractable manner.

**Strengths:**

* Tractable inference in DGNs: the tractable assumption over input distributions and SPN graphical properties enforcement allows for tractable probabilistic inference. This feature allows for a sound way of dealing with missing data.
* Probabilistic modeling of distributions over vertexes is beneficial in some specific applications, as it seems to be the case in the chemical domain. These are encouraging results for graph-based solutions in downstream tasks.
* Throughout and convincing experiments while exploiting well-established deep learning techniques such as residual connections.

**Weaknesses:**

* The manuscript would benefit from a theoretical discussion on the implications of generating tree SPNs from induced graphs, for instance, when capturing cyclic information.
* The paper does a good job motivating the work from the DGN perspective by bringing tractable probabilistic inference capabilities. However, the manuscript's relevance could be improved by highlighting the other way around: novel theoretical results to SPNs.
* Please fix the "Scarce Supervision" paragraph under Section 6: it currently contains a "Lorem ipsum" placeholder.

**Questions:**

* How do imputation methods compare with the formal way of dealing with missing data through probabilistic inference?
* Could you comment on the empirical convergence of the model? Was the model susceptible to variations on parameter initialization? And how did hyper-parameters were tuned?

---

> ### Author Response · Authors · 2023-11-15
> **Response to Reviewer KgLo**
>
> Thank you for the positive assessment of our work. Below we address the comments and questions posed by the reviewer.
>
> **Q:** The manuscript would benefit from a theoretical discussion on the implications of generating tree SPNs from induced graphs, for instance, when capturing cyclic information.
>
> **A:** In Section 4.4, we discuss the theoretical implications of generating a hierarchy of SPNs from cyclic graphs; in particular, the pseudo-likelihood might result in over-optimistic behavior in seriously degenerate cases (e.g., a dataset of graphs that are made of self-loops only). Note that all message-passing architectures, including GSPNs, rely on some approximation of the mutual dependencies induced by cycles in the graph.
>
> **Q:** The paper does a good job motivating the work from the DGN perspective by bringing tractable probabilistic inference capabilities. However, the manuscript's relevance could be improved by highlighting the other way around: novel theoretical results to SPNs.
>
> **A:** This is a great suggestion, thank you. We have highlighted the relevance of our contribution in terms of the SPN literature, namely how to build SPNs as the composition of simpler ones to tackle problems in more complex domains, at the end of the Introduction section.
>
> **Q:** Please fix the "Scarce Supervision" paragraph under Section 6: it currently contains a "Lorem ipsum" placeholder.
>
> **A:** We have fixed the placeholder, apologies about this.
>
> ### Questions:
>
> **Q:** How do imputation methods compare with the formal way of dealing with missing data through probabilistic inference?
>
> **A:** It is possible that we did not fully understand the question. To the best of our knowledge, there is no formal way of dealing with missing data in the context of topologically varying graphs (that is, Markov Networks are not an option here). In particular, GSPN is the first proposal to model missingness while conditioning on the graph structure. Dealing with missingness without conditioning on the graph leads to a degradation of performances as reported in Table 2 (Gaussian/GMM baselines).
> From a more technical standpoint, the major difference compared to classical imputation methods is that we adopt classical probabilistic inference from missing data at the local level of a node in the tree, but we propose a probabilistically tractable way to propagate the information of missingness globally across the graph.
>
> **Q:** Could you comment on the empirical convergence of the model? Was the model susceptible to variations on parameter initialization? And how did hyper-parameters were tuned?
>
> **A:** Regarding convergence, we did observe some light instability when training with higher learning rates, meaning sudden bumps in the training curves that were quickly compensated, but that did not affect our result in the end. As discussed in Section 5 (please refer to the paper for more details), we tuned hyper-parameters automatically with a grid search, and we selected the best configuration on the validation set. Such configuration is then re-trained and eventually evaluated on the test set, which is never seen, directly or indirectly, by the model.
>
> --
>
>
> We hope to have adequately addressed the questions and concerns of the reviewer so that the final score might be increased. Many thanks again for helping us strengthen our contribution.

---

### Official Review · Reviewer_ReHv · 2023-11-07

**Soundness:** 3 good
**Presentation:** 3 good
**Contribution:** 3 good
**Rating:** 5
**Confidence:** 3

**Summary:**

While sum-product networks have been well-studied and proven to be efficient in tractable learning and answering probabilistic queries, all previous studies focus on data in the standard forms (e.g. numerical values or discrete classes). However, it has not been well-studied for graph representation learning and related areas. This paper introduces a new probabilistic framework Graph-Induced Sum-Product Networks (GSPNs), which achieves efficiency and efficacy by utilizing hierarchies of SPNs that allow transferable parameters. Extensive experiments are conducted to analyze the role of hyper-parameters and the model's ability to answer probabilisitic queries.

**Strengths:**

This paper studies an interesting and important problem, which is probabilistic queries for graph learning. Like standard sum-product networks, GSPNs can properly marginalize out missing data in graphs and provide more interpretability, in contrast to deep graph neural networks.

The construction of the networks (page 4 to 6) is detailed, and the hierarchical relationship is well-described. The optimization objective (Equation 1) is expected.

Section 4.2 emphasizes a major advantage of GSPNs, or probabilistic circuits in general, which is the ability to infer with incomplete data. The content in the section also provides the justifications on why certain operations are chosen (summing the average predictions of each mixture in the top SPN).

**Weaknesses:**

The writing of the beginning of Chapter 4 and Section 4.1 could be improved to a reasonable extent. The construction of the tree is highly technical and such a compact text makes the understanding difficult. The authors may consider the following two improvements: 1) write the process in a more rigorous way like a mathematics or TCS paper, i.e. formal definitions of the function $m(\cdot)$ and the heights $\ell$ of the tree; 2) add more figures to illustrate the construction process.

The subject of sum-product networks has a rich theoretical background, while this paper has little theoretical justifications, unless I missed anything. Many operations (such as the construction of the tree, transforming parameters, and summing top SPNs for inference with incomplete data in Section 4.2) are only justified in the hand-wavy way. Although ML is a highly empirical subject, probabilistic circuits are involved for interpretable inference and therefore, a reasonable amount of theoretical justifications may be necessary.

**Questions:**

1. Please refer to the second paragraph in the weakness section in case I missed any substantial theoretical justifications.

2. For Equation 1 on page 4, what exactly is the function $m_v(\cdot)$? Also, since $n_1$ is the root, it makes sense so that all other nodes are conditioned. However, when we infer other nodes, e.g. $m_v(n_2)$, do we try to optimize $\log \prod_{v \in V} P_{\Theta} (x_{m_v(n_2)} | x_{m_v(n_2)}, \cdots ) $?

3. In Section 4.2, how is the equation $x_v^{mis} = \sum_{i}^{C} \mathbb{E} [X_1^{mis} | Q_1^L = i] \times h_1^L(i) $ derived?

---

> ### Author Response · Authors · 2023-11-15
> **Response to Reviewer ReHv**
>
> We are grateful for the constructive feedback and the positive assessment of our work. In the following, we provide answers to the points raised, hoping that the reviewer will consider raising the final score as a consequence.
>
> **Q:** The writing of the beginning of Chapter 4 and Section 4.1 could be improved to a reasonable extent. The construction of the tree is highly technical and such a compact text makes the understanding difficult. The authors may consider the following two improvements: 1) write the process in a more rigorous way like a mathematics or TCS paper, i.e. formal definitions of the function m() and the heights \ell of the tree; 2) add more figures to illustrate the construction process.
>
> **A:** We agree with the writing suggestions and we tried our best to change the manuscript accordingly despite the severe space limitations. In particular, we have provided a more formal description of the height of trees and nodes that follows the classical definitions in theoretical computer science, and we better formalized the mapping from nodes of a tree to vertices in the original graph. Accordingly, and due to the impossibility of adding extra figures, we opted for highlighting the recursive construction of the computational trees in Figure 1 and explicitly referred to the mapping $m_v(\cdot)$ in the caption. Thank you for helping us improve this part.
>
> **Q:**: The subject of sum-product networks has a rich theoretical background, while this paper has little theoretical justifications, unless I missed anything. Many operations (such as the construction of the tree, transforming parameters, and summing top SPNs for inference with incomplete data in Section 4.2) are only justified in the hand-wavy way. Although ML is a highly empirical subject, probabilistic circuits are involved for interpretable inference and therefore, a reasonable amount of theoretical justifications may be necessary.
>
> **A:** The reviewer mentions the numerous theoretical results for SPNs, and we indeed make use of these results (locally valid distributions, ability to tractably compute marginals, etc.) in our work. This sets our proposed framework apart from prior work in the graph machine learning literature which does not rely on SPNs and graphical models. What we would like to highlight, in addition, is that the design of GSPN is theoretically motivated, as it builds on architectural biases, taken from the literature, which have well-known theoretical foundations [*,**]. The objective of our current work is to propose, for the first time and to the best of our knowledge, a new class of models integrating message-passing NNs and SPNs, and not to derive new theoretical results about SPNs. We agree, however, that this is interesting future work.
>
> ### Questions:
> 1. Please read above.
> 2. Thank you for spotting this: the text was ambiguous, as it talked about the optimization of a single computational tree while Equation 1 shows how to optimize the pseudo log-likelihood for an entire graph $g$. We have corrected the text accordingly in the revised version.
> 3. There is no derivation involved in the equation $\textbf{x}_v^{mis} = \sum_i^C \mathbb{E}\big[\textbf{X}_1^{mis} \mid Q^{L}_1=i\big] * \textbf{h}^L_1(i)$: because the SPN of the root node is a Naive Bayes model, which is a mixture model, we just applied the conditional mean imputation strategy defined in Zio et al. (2007) for finite mixture models to our context. We have clarified this in the updated manuscript.
>
>
>
> [*] Micheli, Alessio. "Neural network for graphs: A contextual constructive approach." IEEE Transactions on Neural Networks 20.3 (2009): 498-511.
>
> [**] Scarselli, Franco, et al. "The graph neural network model." IEEE transactions on neural networks 20.1 (2008): 61-80.
>
> --
>
>
> We hope to have addressed the concerns of the reviewer, especially those regarding the objectives of our work. We remain available should further clarifications be needed, and we thank the reviewer again for the valuable feedback.

---

> > ### Comment · Reviewer_ReHv · 2023-11-23
> > **Response to the authors**
> >
> > I appreciate the authors' feedback. I highly advocate the novelty of this work, but since the weaknesses are not addressed at this moment, I will keep my original assessment.

---

### Author Response · Authors · 2023-11-20
**General Response to Reviewers**

We want to thank the reviewers again for the positive assessment of our work and for providing suggestions for improvement; we have revised the manuscript accordingly. It would be very much appreciated if you could acknowledge our modifications and let us know if there is anything else we can address.

We would like to stress once more that our work constitutes one first step to apply SPNs to the context of message-passing, and it is motivated by theoretical results in the field of graph machine learning. Thank you again for your help in improving the manuscript.

---

### Meta-Review · Area_Chair_TXGy · 2023-12-08

**Metareview:**

First of all, I'd like to assure the authors that the first review is not taken into account in the final decision. I read the paper myself, and in my opinion, it is an interesting and novel paper in combining and using SPNs/PCs in graphs. Due to the nice properties of SPNs, the proposed framework allows tractable inference over graphs, which can also deal with missing data that appear on the vertices of a graph. The authors also did comprehensive experiments to showcase the application of the proposed framework.

During the rebuttal period, the authors addressed the concerns of the reviewers, and I believe that the paper is ready for publication. I recommend acceptance.

**Justification For Why Not Higher Score:**

The work is a nice application of SPNs over graph-structured data, while interesting in the specific applications, the paper itself does not contribute fundamentally to the study of SPNs themselves, as also pointed out by R2.

**Justification For Why Not Lower Score:**

The authors have addressed all the concerns during rebuttal and the paper is technically sound.

---

### Decision · Program_Chairs · 2024-01-16

Accept (poster)